# Influenza A Virus in Pigs in Senegal and Risk Assessment of Avian Influenza Virus (AIV) Emergence and Transmission to Human

**DOI:** 10.3390/microorganisms11081961

**Published:** 2023-07-31

**Authors:** Mamadou Malado Jallow, Mamadou Aliou Barry, Amary Fall, Ndiendé Koba Ndiaye, Davy Kiori, Sara Sy, Déborah Goudiaby, Mbayame Ndiaye Niang, Gamou Fall, Malick Fall, Ndongo Dia

**Affiliations:** 1Institut Pasteur de Dakar, Département de Virologie, Dakar BP 220, Senegal; mamadoumalado.jallow@pasteur.sn (M.M.J.); amary022@hotmail.com (A.F.); ndiendekoban@gmail.com (N.K.N.); davy.kiori@pasteur.sn (D.K.); sara.sy@pasteur.sn (S.S.); deborah.goudiaby@pasteur.sn (D.G.); mbndiaye2002@gmail.com (M.N.N.); gamou.fall@pasteur.sn (G.F.); 2Département de Biologie Animale, Faculté des Sciences et Techniques, Université Cheikh Anta DIOP de Dakar, Dakar BP 206, Senegal; malickfal@yahoo.fr; 3Institut Pasteur de Dakar, Unité d’Epidémiologie des Maladies Infectieuses, Dakar BP 220, Senegal; aliou.barry@pasteur.sn

**Keywords:** influenza, surveillance, pig, Senegal, epidemiology, serological, avian

## Abstract

We conducted an active influenza surveillance in the single pig slaughterhouse in Dakar to investigate the epidemiology and genetic characteristics of influenza A viruses (IAVs) and to provide serologic evidence of avian influenza virus (AIV) infection in pigs at interfaces with human populations in Senegal. Nasal swab and blood samples were collected on a weekly basis from the same animal immediately after slaughter. Influenza A viruses were diagnosed using RT-qPCR and a subset of positive samples for H3 and H1 subtypes were selected for full genome amplification and NGS sequencing. Serum samples were tested by HI assay for the detection of antibodies recognizing four AIVs, including H9N2, H5N1, H7N7 and H5N2. Between September 2018 and December 2019, 1691 swine nasal swabs were collected and tested. Influenza A virus was detected in 30.7% (520/1691), and A/H1N1pdm09 virus was the most commonly identified subtype with 38.07% (198/520), followed by A/H1N2 (16.3%) and A/H3N2 (5.2%). Year-round influenza activity was noted in pigs, with the highest incidence between June and September. Phylogenetic analyses revealed that the IAVs were closely related to human IAV strains belonging to A/H1N1pdm09 and seasonal H3N2 lineages. Genetic analysis revealed that Senegalese strains possessed several key amino acid changes, including D204 and N241D in the receptor binding site, S31N in the M2 gene and P560S in the PA protein. Serological analyses revealed that 83.5% (95%CI = 81.6–85.3) of the 1636 sera tested were positive for the presence of antibodies against either H9N2, H5N1, H7N7 or H5N2. Influenza H7N7 (54.3%) and H9N2 (53.6%) were the dominant avian subtypes detected in Senegalese pigs. Given the co-circulation of multiple subtypes of influenza viruses among Senegalese pigs, the potential exists for the emergence of new hybrid viruses of unpredictable zoonotic and pandemic potential in the future.

## 1. Introduction

Influenza, also known as the flu, is a respiratory illness affecting around 1 billion individuals per year, with 3 to 5 million severe cases and 290,000 to 650,000 deaths worldwide [1]. It is caused by viruses belonging to the *Orthomyxoviridae* family, which consists of segmented, single-stranded, negative-sense RNA viruses [2]. This family contains four influenza virus genera: *Alphainfluenzavirus* (influenza A virus, IAV), *Betainfluenzavirus* (influenza B virus, IBV), *Gammainfluenzavirus* (influenza C virus, ICV) and *Deltainfluenzavirus* (influenza D virus, IDV), which are classified according to antigenic variations of their nucleoprotein (NP) and matrix 1 (M1) proteins [3,4,5]. Infections with human IAV and IBV are responsible for annual seasonal epidemics leading to about half a million deaths worldwide [6]. Influenza A viruses (IAV), which emerge periodically from animals, also cause occasional pandemics in the human population, leading to a disastrous impact on global human health with elevated mortality rates [7,8]. In contrast, ICV infections are milder and IDV, which infect pigs and cattle, have not been associated with human disease [9,10]. IAV subtypes are distinguished according to genetic and antigenic characteristics of their surface glycoproteins, hemagglutinin (HA) and neuraminidase (NA) [11]. To date, 18 known HA (H1–H18) and 11 NA (N1–N11) subtypes of IAVs have been isolated from aquatic birds, which are therefore considered the natural reservoirs of IAVs [4,12].

Influenza A viruses pose a significant risk to human health because of their wide host range (swine, birds, horses, dogs, cats, sea mammals, etc.) and capacity to reassort (mechanism by which novel influenza viruses can be generated). This occurs when two or more different influenza virus strains infect the same host cell and exchange genetic material, resulting in the creation of a new virus with a combination of genes from the parent strains) into novel viruses (due to the nature of its segmented genome) that can cause serious epidemics or pandemic [13]. Therefore, monitoring circulating influenza strains in animals and assessing their potential to cross the species barrier and cause infections in humans are important for good pandemic preparedness. This process of crossing the species barrier can occur either directly or after reassortment that results in the generation of new strains capable of replicating efficiently in humans, despite having a distinct antigenic profile. Birds (wild and domestic) and pigs play a crucial role in this genetic dynamic [14]. Pigs are an important host in IAV ecology and are widely considered as a “mixing vessel”. They appeared unique in having both α-2,3-linked sialic acid (α-2,3-SA) and α-2,6-linked sialic acid (α-2,6-SA) receptors distributed throughout their respiratory tracts, where avian, swine and human influenza viruses can undergo genetic reassortment (when they infect the same host cell and exchange genetic material), resulting in the creation of novel viruses [15,16,17]. Once established in swine, novel reassortant viruses could pose a substantial threat to swine populations and, with introductions and/or reintroductions into humans, could potentially pose outbreak and pandemic risks [17]. This risk is exemplified by the emergence of the swine-origin A/H1N1pdm09 virus in Mexico and the United States in April 2009 [18,19]. This virus, resulting from reassortment in pigs, rapidly spread throughout the human population, causing the first global influenza pandemic of the 21st century [20], ultimately affecting 214 countries and regions with nearly 19,000 deaths in the first 14 months since its recognition in 2009 [15]. Recent human infections with swine H3N2 variants (H3N2v) have been another result of IAV reassortment in pigs [21,22]. Thus, since the emergence of the H1N1 2009 pandemic influenza virus A(H1N1)pdm09, surveillance of swine influenza (SI) has been intensified worldwide [23,24,25].

In Senegal, over the past decade, pig and poultry rearing have increased dramatically, and they are often raised together in low biosecurity settings, posing a risk for influenza reassortments in pigs. In 2020, Senegal experienced its first documented outbreak of the high pathogenicity avian influenza virus (HPAIV) H5N1 in a poultry farm having a total of 102,000 birds with a mortality rate of 58% [26], and the first human case of avian H9N2 infection was reported [27]. At the moment we are writing this document, Senegal is facing a large outbreak of HPAIV H5N1 in the north with high mortalities reported in wild birds. These events sound like an alert to a possible circulation of avian influenza virus (AIV) in the Senegalese community (animal and human). Senegal has a long experience of influenza surveillance, but it has exclusively focused on humans. Thus, the lack of surveillance at the human–animal interface is the real weakness of influenza surveillance in Senegal. Given the complexity of influenza virus ecology, a “One health” surveillance approach is essential to monitor, in real time, the evolution and transmission dynamics of influenza viruses. So, here, we conducted an active virological surveillance in the single pig slaughterhouse in Dakar between September 2018 and December 2019 to investigate the epidemiology and genetic characteristics of IAVs, as well as serological evidence of four avian influenza virus infections (H9N2, H5N1, H7N7 and H5N2) in pigs.

## 2. Materials and Methods

### 2.1. Surveillance and Sample Collection

Active swine influenza surveillance in the single pig slaughterhouse in Dakar (capital city) was performed through the National Influenza Center (NIC) activities. It is a unique place where pigs from different regions of Senegal are temporarily fattened before being slaughtered for the sale of meat (Figure 1). From September 2018 to December 2019, blood and nasal swabs were collected on randomly selected pigs immediately after slaughter. About 5 mL of blood was collected into labelled sample bottles without anticoagulants and allowed to clot. Nasal swabs were placed in 2 mL of universal viral transport medium (VTM, Becton Dickinson and company, Milano, Italy) and transported at a controlled temperature (4 °C) to Institut Pasteur de Dakar (IPD) the same day. Upon arrival in the laboratory, swab samples were immediately processed for influenza screening and identification. Aliquots of each sample were also stored at −80 °C for further studies. With regard to blood specimens, serum was extracted from clotted blood samples via centrifugation for 10 min at 3000 rpm and stored in new labelled cryotubes at −20 °C until tested for antibodies against AIVs. Data on region of origin and date of samples collection were recorded for each sampled pig.

### 2.2. Detection of Swine Influenza A Viruses (swIAVs) by RT-qPCR

#### 2.2.1. RNA Extraction from Pigs’ Nasal Swabs

Ribonucleic acid (RNA) was extracted from 200 μL of VTM fluid containing pig nasal swabs using the QIAamp Viral RNA kit (QIAGEN, Valencia, CA, USA) as per the manufacturer’s instructions. Each RNA sample was eluted in a final volume of 60 µL and immediately used for the screening of influenza viruses or kept at −80 °C until use.

#### 2.2.2. Screening of Influenza Viruses

Reverse transcription polymerase chain reaction (RT-qPCR) was performed to initially screen for all IAVs, using specific primers and probe targeting a conserved region of the influenza matrix gene, and provided by the CDC International Reagent Resources (IRR) using an ABI 7500 device, according to the CDC procedure [28]. Briefly, tests were performed in singleplex using the AgPath-ID™ One-Step RT-PCR kit (Ambion, Foster City, CA, USA). For each sample, real-time PCR was carried out in a total reaction volume of 25 μL consisting of 5 μL Nuclease-free water, 0.5 μL of each primer (diluted at 10 μM), 0.5 μL of probe (diluted at 10 μM), 12.5 μL of 2X RT-PCR Buffer, 1 μL 25X RT-PCR Enzyme Mix and 5 μL of RNA template under the following cycling conditions: reverse transcription step of 15 min at 50 °C, initial denaturation step of 3 min at 95 °C, followed by 45 PCR cycles of 15 s at 95 °C and 30 s at 55 °C. For each test, positive (provided by IRR) and negative (nuclease-free water) controls were included for validation. A test was considered valid when there was amplification of the positive control and absence of amplification of the negative control (to rule out any contamination during the testing process). Subsequently, all M gene-positive samples (Ct-values < 36) were subtyped via RT-qPCR assays using specific primers and probe that specifically amplify and discriminate the HA and NA genes from the different lineages of swine IAVs known to be enzootic in pig herds (H1N2, H3N2 and H1N1pdm09). The same condition as the initial screening was used for subtyping.

### 2.3. Complete IAV Genome Amplification and NGS Sequencing

#### 2.3.1. cDNA Synthesis

A subset of positive samples for H3 and H1 subtypes were selected for full genome amplification and NGS sequencing. Viral RNA was initially extracted from positive nasal swabs and reverse transcription carried out using a RevertAid First Strand cDNA Synthesis Kit (Thermo Scientific, Vilnius, Lithuania) and the universal primer Uni12 (AGCAAAAGCAGG). Briefly, for each sample, a mixture of 15 μL of RNA and 2 μL of Uni12 (10 μM) primer was initially incubated at 65 °C for 5 min and put in ice for 1 min. Then, the following constituents, including 6 μL of 5X Reaction Buffer, 1.5 μL of RiboLock RNase Inhibitor (RI), 1.5 μL of RevertAid Reverse Transcriptase (RT) and 3 μL of dNTP Mix (10 μM), were added into each tube. The reaction was carried out at 42 °C for 1 h and was terminated by heating at 70 °C for 5 min.

#### 2.3.2. Multi-Segment PCR of Swine Influenza A Virus (swIAV)

To characterize swIAV using next-generation sequencing, complete genomes (all eight-gene segments) were amplified through a single conventional PCR run by using the LongAmp Tag 2X Master Mix kit (New England Biolabs, Ipswich, MA, USA) with primers MBTuni-12 (5′-ACGCGTGATCAGCAAAAGCAGG-3′) and MBTuni-13 (5′-ACGCGTGATCAGTAGAAACAAGG-3′) as previously reported by Zhou et al. [29]. Briefly, for each sample, PCR amplification was carried out in a total reaction volume of 50 μL consisting of 11 μL Nuclease-free water, 2 μL of each primer (diluted at 10 μM), 25 μL of Master mix and 10 μL of cDNA template under the following thermal cycling conditions: initial denaturation at 94 °C for 30 s; five cycles of 94 °C for 30 s, 45 °C for 30 s and 65 °C for 3 min; followed by 45 PCR cycles of 30 s at 94 °C, 30 s at 56 °C, 3 min at 65 °C; and a final extension of 65 °C for 10 min. PCR products were analyzed in a 1% agarose gel stained with ethidium bromide, using 1xTAE as the electrophoresis running buffer. Bands containing amplicons were cut and purified with the NucleoSpin^®^ Gel and PCR Clean-Up (Macherey-Nagel GmbH & Co. KG, Düren, Germany) according to the supplier’s protocol. After amplicons purification, products were quantitated and normalized to approximately 100 ng using the Qubit fluorimeter (Invitogen life technologies), and then sent to the Platforme de Microbiologie Mutualisée (P2M) hosted at the Institut Pasteur de Paris for NGS sequencing.

#### 2.3.3. Next Generation Sequencing

For whole genome sequencing of swIAVs, an amplicon-based next-generation sequencing approach was used. Briefly, the pooled PCR products underwent bead-based tagmentation using the Nextera DNA Flex library preparation kit. The adapter-tagged amplicons were cleaned-up using AmpureXP purification beads (Beckman Coulter, High Wycombe, UK) and amplified using one round of PCR. The PCRs were indexed using the Nextera CD indexes (Illumina, Inc., San Diego, CA, USA) according to the manufacturer’s instructions. The obtained libraries were quantified using a Qubit 4.0 fluorometer (Invitrogen Inc., Waltham, MA, USA), using the Qubit dsDNA High Sensitivity assay according to the manufacturer’s instructions. The pooled libraries were further normalized to 1 nM concentration, and 5 μL of each normalized pool containing unique index adapter sets were combined in a new tube. The final library pool was denatured with 0.1 N sodium hydroxide, and the 12 pM sample library was spiked with 10% PhiX. Libraries were loaded onto an Illumina MiSeq platform with 151 bp paired-end reads using the Miseq reagents kit v3, according to the manufacturer’s protocol (Illumina, San Diego, CA, USA).

To generate the consensus genomes of swIAVs, raw reads from Illumina sequencing in FASTQ format were assembled via de novo assembling using the web-based pipeline Genome Detective (https://www.genomedetective.com/, accessed on 5 November 2019).

### 2.4. Phylogenetic Analysis

For phylogenetic analysis, the newly generated sequences were combined with reference sequences retrieved from the National Center for Biotechnology Information (NCBI, https://www.ncbi.nlm.nih.gov, accessed on 6 November 2019) and from the Global Initiative on Sharing All Influenza Data (GISAID, http://www.gisaid.org, accessed on 6 November 2019), which represent the range of genetic diversity of swIAV worldwide. The eight datasets corresponding to the eight gene segments of IAV were initially aligned using MAFFT v7.450 software [30] and then adjusted manually in Bioedit v7.1.3.0 [31]. Maximum likelihood phylogenetic trees with a general time reversible model (GTR) were inferred using MEGA 7 software, and the reliability of tree topologies was assessed via bootstrap analysis with 1000 replications [32]. Only bootstrap replicates with values ≥70 were kept on the trees.

### 2.5. Hemagglutination Inhibition (HI) Assays of Serum Samples

The serum samples were tested via HI assay for the detection of antibodies recognizing four reference viruses, A/mallard/Netherlands/12/2000 (H7N7), A/Chiken/Hong Kong/G9/1997 (H9N2), A/Egypte/321/2007 (H5N1) and A/pheasant/New Jersey/1355/1998 (H5N2), provided by the International Reagent Resource (https://www.internationalreagentresource.org, accessed on 2 February 2021). Prior to conducting the assays, samples and positive sera control were treated with receptor-destroying enzymes (RDE; Denka Seiken Co. Ltd., Tokyo, Japan) to remove non-specific hemagglutination inhibitors. Briefly, 3 volumes of RDE (reconstituted with 25 mL of 0.85% physiological saline solution) added to 1 volume of serum were incubated in a water-bath at 37 °C overnight, followed by heat inactivation in a water-bath the next morning at 56 °C for 30 min to inactivate any remaining RDE. Six volumes of 0.85% physiological saline were then added to RDE-treated sera to obtain a final dilution of 1:10.

The HI tests were subsequently performed using a 96-well microtiter plate following WHO recommendations for animal influenza diagnosis [33]. Dilutions which contained 8 HA units/50 µL of reference antigens were obtained before each test and a back-titration of the 8 HA was performed to verify its correctness. Briefly, 25 µL of phosphate-buffered saline (PBS) was added to wells 2 through 12 of each row prior to the addition of 50 µL of each treated serum to the first well labelled for it. Serial twofold dilutions were made by transferring 25 µL amounts from the first row to successive rows, and the last 25 µL was discarded after well 10 to give a dilution of 1:20 through 1:5120. The last two columns were used as red blood cells (RBC) and antigens control wells. Twenty-five microliters of antigen containing 8HA units/50 µL of the reference virus was then added to the appropriate wells. Plates were agitated manually and incubated at room temperature for 30 min to allow potential antigen–antibody reaction, after which 50 µL of 0.75% suspension of human type O RBCs was added to all wells. Plates were manually agitated and incubated at room temperature for 1 h and observed for HI reactions. This was performed by tilting and observing the presence of tear-shaped streaming at the same rate as the control wells containing RBCs only. The HI antibody titer of each sample was determined as the reciprocal of highest serum dilution that completely inhibited hemagglutination of 8 HA units of the virus. Considering previous swine influenza virus serosurveys [16,34,35], a sample was considered seropositive if it had HI titer ≥ 1:80, as lower titers may be due to non-specific reactions.

### 2.6. Statistical Analysis

All data obtained from this active influenza surveillance, including the laboratory results, were compiled in an Excel spreadsheet. Subsequent data analysis was performed in R Statistical Software (R.3.0.1 version), using chi-square (χ2) and Fisher’s exact tests to support the comparisons of the categorical data, where a *p*-value < 0.05 was regarded statistically significant. The proportions were reported with 95% confidence intervals (CI).

### 2.7. Ethical Statement and Permission

Permission was sought from the swine-owners and swine workers at the slaughterhouse where the samples were collected. All procedures performed in studies involving animal samples were in accordance with the international, national and/or institutional guidelines for the care of animals.

## 3. Results

### 3.1. Sample Collection

From 26 September 2018, through 31 December 2019, a total of 1691 nasal swabs and 1636 blood samples were collected from pigs originating from eight out of the fourteen administrative regions of Senegal, of which 75.7% (1281/1691) were registered in 2019. All pigs slaughtered in the slaughterhouse were adults. The majority of samples were collected from pigs originating from the region of Fatick, with nearly half of the overall collected specimens (46%; 778/1691), followed by pigs imported from the Thiés region (26.5%; 449/1691), Diourbel (6.4%; 109/1691) and Dakar (4.1%; 69/1691), while only 0.9% (15/1691), 1.2% (20/1691), 2.5% (43/1691) and 2.6% (44/1691) were collected from Kaffrine, Kaolack, Louga and Saint-Louis region pigs, respectively (Table 1). The origin of 9.7% (164/1691) of the sampled pigs was unknown.

### 3.2. Detection of IAVs

Overall, of the 1691 swine nasal swabs tested by RT-qPCR, 30.7% (520/1691) yielded positive results for IAV. From the 520 IAV-positive samples that were subtyped over the study period, A/H1N1pdm09 virus was the most commonly identified, accounting for 38.1% (198/520), followed by A/H1N2 and A/H3N2 viruses, with detection rates of 16.3% (85/520) and 5.2% (27/520), respectively (Table 1). For 275 (51.9%) IAV-positive samples, the subtype could not be determined due to exhibiting M gene Ct-values > 33, which indicated an apparent very low viral load. Co-infections of several subtypes including H1N1pdm09/H1N2 and H1N1pdm09/H3N2 were encountered in 13.5% (70/520) and 1.5% (08/520), respectively. Five triple infections (A/H1N1pdm09/A/H1N2/A/H3N2) were also noted.

### 3.3. Circulation Pattern of Swine Influenza A Viruses (swIAVs)

Overall, we observed a similar trend for A/H1N1pdm09 and A/H1N2 infections which occurred throughout the surveillance period. In 2018, the weeks 46 to 51 (November and December) demonstrated the highest number of influenza infections among pigs, while in 2019, an overall increase in the number of A/H1N1pdm09 detection (which was the most prevalent subtype) was registered from week 25 to week 39 (June–September), when the seasonal epidemic occurred in humans, with a peak of detection in weeks 25 and 26 (June). The few cases of A/H3N2 detected in this study was encountered in week 49 (December) of 2018 and weeks 10, 17, 39 and the last trimester (week 43–52) of 2019 (Figure 2).

### 3.4. Phylogenetic Analysis of IAV Strains Circulating in Senegalese Pigs, 2018–2019

A total of 14 IAVs that circulated among Senegalese pigs between 2018 and 2019, including eleven H1 and three H3 viruses, were selected to generate 14 complete genomes (3 in 2018 and 11 in 2019). All sequences generated in the course of the present study were preliminarily analyzed, performing a nucleotide query step in the Genbank database (http://blast.ncbi.nlm.nih.gov/BLAST; accessed on 6 November 2019) and identifying closely related sequences. Results of the BLAST searches revealed that all eight RNA segments (PB2, PB1, PA, HA, NP, NA, M and NS) of the eleven H1 subtypes, were most closely related to the 2009 pandemic influenza A (H1N1) virus that circulated among human populations throughout the world, and the eight gene segments of the three H3 swine influenza possessed high nucleotide sequences similarities to the human seasonal H3N2 viruses.

We undertook phylogenetic analysis to ascertain more precisely the genetic relationships among the A/H1N1 and A/H3N2 viruses detected in this study and their putative closest ancestors. All H1 viruses fully sequenced in this study clustered closely with their counterparts circulating among Senegalese human population, and they all belonged to the H1N1pdm09-like sublineage of the 1A classical swine lineage in all eight gene segments (Figure 3A). Phylogenetic analysis of the HA gene of Senegalese strains showed that the three isolates were closely related to the contemporary seasonal H3N2 viruses circulating in the Senegalese human population and belonged to the cluster II lineage (Figure 3B). The other genes (NA, PB2, PB1, PA, NP, M and NS) were also related to the corresponding genes of H3N2 human influenza viruses.

### 3.5. Genetic Analysis of Amino Acid Residues of Senegalese Strains

The deduced amino acids of the individual sequences obtained in the present study were analyzed to identify amino acids that may be involved in virus virulence, pathogenicity, mammalian transmission, receptor-binding specificity and antiviral drugs resistance. The FluServer (https://flusurver.bii.a-star.edu.sg/, accessed on 14 December 2022) was used to identify key amino acid mutations. The HA cleavage sites of all H1 viruses were characterized as monobasic residue PSIQSR/GLF, while those of the H3 subtype contained the motif PEKQTR/GI at the cleavage site, which is consistent with the characteristics of low pathogenic influenza viruses [17]. Compared with reference strains A/GuangdongMaonan/SWL1536/2019 (H1N1) and A/Brisbane/02/2018 (H1N1), HA of Senegalese H1 strains had several amino acid substitutions, including V204D, E206Q in 10 isolates and R240Q in 1 isolate. Equivalent mutations at these positions have been shown to have an increasing affinity of receptor-binding to SA_2,6Gal. For H3N2 viruses, mutations F153S, S154A and N241D in the HA protein were encountered in all three isolates, indicating preferential binding to the human receptor as well. Mutations H274Y and N294S, which are the NA mutations most frequently associated with oseltamivir resistance, were not detected in any of the Senegalese strains. The S31N mutation in the M2 gene, which contains the key amino acid target of amantadine drugs, was encountered in all Senegalese strains. For internal genes, one virus (A/swine/SEN/0096/2018) exhibited P560S amino acid substitution in the PA protein, whereas NS1-M106V and NS2-M19L mutations were encountered in two (A/swine/SEN/1174/2019/ and A/swine/SEN/0096/2018) viruses (Table 2).

### 3.6. Serological Evidence of AIV Infection in Pigs

To investigate avian influenza virus (AIV) infection in pigs from Senegal, HI activity was tested from serum samples collected during the surveillance period by using four different AIVs as antigens, namely A/H9N2, A/H5N1, A/H7N7 and A/H5N2. The seroprevalence rates of each virus are summarized in Table 3. Of the 1636 serum samples analyzed with the HI test, 1367 (83.5%; 95%CI = 81.6–85.3) were positive for the presence of antibodies against either A/H9N2, A/H5N1, A/H7N7 or A/H5N2. In serum samples collected in 2018, AIV was detected in 92.4% (364/394), whereas in 2019, HI antibody was detected in 80.4% (1003/1248). Overall, influenza A/H7N7 and A/H9N2 were the dominant avian subtypes detected, with seropositivity rates of 54.3% (889/1636) and 53.6% (877/1636), respectively. Antibodies against A/H5N1 and A/H5N2 were detected at similar rates, with 31% (506/1636) and 30.7% (502/1636), respectively (Figure 4). Among the tested serum samples, we found that some pigs had antibodies against different serotypes simultaneously, with 549 being positive for both A/H9N2 and A/H7N7 (40.2%). A total of 387 samples were copositive for A/H9N2 and A/H5N1 (28.3%), whereas 335 specimens were copositive for A/H7N7 and A/H5N1 (24.5%). Antibodies against all four AIVs targeted in this study were detected simultaneously in 181 individual pigs (13.2%).

### 3.7. Geographical Distribution of AIV Seropositive Cases

In order to understand the range expansion of AIV infection in pigs in Senegal, the geographical distribution was explored and shown in Figure 5. All serum samples were collected from pigs imported from eight regions of Senegal, and the positivity rates were different between regions. The regions of Fatick (38%; 621/1636) and Thiés (22.5%; 368/1636), where the domestic pigs breeding density was high, had the highest seropositivity rates, followed by the region of Diourbel, located in the western part of the country, and the capital city Dakar, with HI seroprevalences of 5.1% (83/1636) and 3.4% (55/1636), respectively. The remaining regions had seropositivity rates ranging from 2.6% (44/1636) in Saint-Louis and Louga (43/1636), located in the northern part of the country, to 1.2% (20/1636) in Kaolack, located in the center of Senegal. In 2018, we detected antibodies against all AIVs targeted in this study in all regions except Kaolack, where we found evidence of infection only for A/H9N2 and A/H7N7 in pigs. However, in 2019, antibodies against all four AIV subtypes were detected in all eight regions.

## 4. Discussion

Movement of hogs during husbandry, trade or marketing for slaughter provides opportunities for transfer and genetic reassortment of IAVs. The latter circulate and evolve continually within swine populations and represent a reservoir of potential human pandemic influenza strains [36]. The emergence of the human 2009 pandemic H1N1 (H1N1pdm09) virus from swine populations refocused public and scientific attention on swine as an important source of IAVs bearing zoonotic potential and underlined the necessity of a continuous global surveillance of swIAVs [11,37]. Whilst much is known about the epidemiology of influenza in humans in Senegal [1,38,39,40], our understanding about the real situation and the genetic diversity of IAVs circulating in swine is largely unknown, hampering control measures and pandemic risk assessment. To fill this knowledge gap, we conducted an active influenza surveillance in the single pig slaughterhouse in Dakar to illustrate the current situation of IAVs in swine populations in Senegal and to provide serologic evidence of AIV infection in pigs. Indeed, over 95% of pigs slaughtered at the abattoir of Dakar are imported from different regions in Senegal, and to the best of our knowledge, no vaccination program for influenza virus has been carried out in this pig population. Our surveillance results should therefore reflect the natural situation of influenza virus infection in the swine populations of these eight regions.

Between September 2018 and December 2019, 30.7% of a total of 1691 nasal swabs collected from pigs tested positive for IAV. This rate was similar to that reported in a study on influenza infection in pigs through a passive surveillance program conducted in 14 European countries from 2010 to 2013, with authors reporting an IAV detection rate of 31% [24]. However, the overall prevalence obtained in this surveillance study was higher compared to those reported in previous studies in several countries, including China with 2% [41], Nigeria with 13.7% [23], Cambodia with 1.5% [42] and Colombia with 13.4% [43]. Furthermore, this report is consistent with previous investigations showing a concurrent circulation of all three swIAV subtypes, including A/H1N2, A/H1N1pdm09 and A/H3N2, in the swine populations in Senegal, but with very different levels of incidence [39,41,44]. The concomitant circulation of many different virus strains in herds increases the risk for co-infections and enables subsequent genetic reassortment, yielding viral progeny with unknown characteristics [45]. As has been reported by several authors around the world [46,47,48], A/H1N1pdm09 was the most prevalent subtype detected in this study, with 38.1% of the overall positive cases, whereas the A/H3N2 virus circulated at very low rate. Unlike our results, Jung et al. reported A/H3N2 as the dominant subtype in pigs in Korea [49]. Co-infections were common in this study, and the most frequently co-detected viruses were H1N1/H1N2, with 13.5% of the overall positive cases. In agreement with our report, several previous studies have provided evidence of swIAVs co-infection in a natural setting [50,51]; this is the case in Thailand, where researchers demonstrated that co-infection of H3N2, H3N1, H1N1 and H1N2 resulted in the creation of at least 18 different genotypes in a naturally reared pig [52]. The differences in the detection rates reported in other works and in this study indicate that other factors could favor a higher or lower contact of hogs with different IAV subtypes, including the period of the year, environmental characteristics and especially the interaction with other hosts susceptible to infection by IAV, like humans and domestic or wild birds [53].

With regard to the circulation pattern, this active surveillance confirmed year-round viral activity for endemic swIAVs, with an overall increase in infections between June and September. The overall prevalence of IAVs detected in pigs follows the human seasonal influenza pattern that was seen in Senegal during these recent years [40]. Indeed, in Senegal, the seasonality of influenza in the human population is well defined after years of regular flu monitoring [38,40]. A regular circulation of influenza virus during the year with a peak in August (in the middle of the rainy season) was noted [38]. A study, which involved five European countries, also reported year-round influenza virus activity in swine herds, with peaks in December and May [46]. Year-round influenza virus activity in pig populations entails a continuing risk of transmission of swIAV from swine to humans.

As has been seen in other geographical regions [23,54,55,56], the phylogenetic and sequence analysis of the whole genomes of the 14 influenza strains obtained in our study revealed that the external genes of the isolates were assigned to the H1N1pdm09-like sublineage of the 1A classical swine lineage and to the human-like H3N2 lineage (cluster II), displaying high identities with human A(H1N1)pdm09 isolates and human seasonal H3N2 viruses, respectively. Similarly, analysis of the remaining gene segments demonstrated that all the internal genes (PB1, PB2, PA, NP, M and NS genes) were closely related to contemporary H1N1 and H3N2 subtypes circulating in the human population in Senegal. This is consistent with a previous study in Vietnam [44] indicating that all eight gene segments of all H1N1 viruses derived from H1N1pdm09. The similarity of influenza viruses circulating in pigs and humans suggest a possible human-to-swine influenza virus transmission in Senegal.

Reassortment events are well known to contribute to genetic variations and evolution of influenza viruses. If a single cell is concurrently infected with more than one virus, IAVs are able to undergo reassortment (because of the segmented nature of the RNA genomes) via intra- and/or inter-species molecular genetic exchanges that may lead to the generation of reassortant viruses with increased cross-species transmissibility, pathogenicity, and lethality, which could cause a human influenza pandemic [57]. Although reassortments are frequently reported in pigs in several countries, such as Vietnam [58], China [59], France [60] and Canada [61], in our study, according to the sequences analysis, no reassortment was encountered, probably due to non-exhaustive sequencing.

Consistent with previous reports [38,62,63], genetic analysis indicated that Senegalese strains possessed several amino acid changes, including D204 and N241D in the receptor binding site, which is known to confer binding of H1 and H3 viruses to the human SA α2-6 receptors, supporting efficient transmission of these viruses to humans [38]. The virulence of influenza viruses in humans is related to their resistance to the antiviral effects of cytokines, such as interferon (IFN), and the mutations P42S and D92E in the non-structural protein 1 (NS1 protein) can increase resistance to IFN [17,64]. Unlike our results, Song et al. [17] found the P42S mutation in influenza viruses circulating in pigs in China, suggesting that these viruses may promote a greater resistance to these cytokines. Several amino acid substitutions, such as E119G/V, D199G, I223K/R/V, S247N, H275Y and N295S (N1 numbering) in neuraminidase protein subtype 1 (N1 protein), and E119D/V, Q136K, I222L, R292K, N294S and the deletion of 245–248 (N2 numbering) in neuraminidase protein subtype 2 (N2 protein), have been detected in viruses associated with oseltamivir resistance [65]. None of the above amino acid changes were detected in the N1 and N2 proteins of the Senegalese strains, indicating their susceptibility to oseltamivir. On the other hand, the S31N mutation in the matrix protein 2 (M2 gene), which contains the key amino acid target of amantadine drugs, was encountered in all Senegalese strains. Similar findings have been reported by several investigators around the world, including Cao et al. in China [65], Mon et al. in Myanmar [66] and Krumbholz et al. in Germany [67]. In contrast to the results of Ali et al. [50], amino acid substitutions, P560S, M106V and M19L in the polymerase acid (PA), NS1 and NS2 proteins, respectively, were observed in few Senegalese strains. Equivalent mutations at these positions have been reported to be related to the virus virulence [68,69,70].

In addition to the virological surveillance, we performed serological testing for antibodies against AIV infection in swine sera collected in Senegal. Our findings revealed a high AIV seroprevalence rate of 83.5% (95%CI = 81.6–85.3) among Senegalese pigs. This result is different to those reported in apparently healthy pigs in Nigeria (44.4%) by Maseko et al. [71] and in China (4.6%) by Yuan et al. [34]. Our findings regarding A/H7N7 (54.3%) and A/H9N2 (53.6%) as the dominant avian influenza subtypes matched the results reported previously [72]. Contrary to the results of Song et al., where no evidence of naturally occurring H5N1 infection in pigs was found [73], in our study, we registered a relatively high seroprevalence rate of the highly pathogenic avian influenza A/H5N1 in Senegalese pigs. Furthermore, we also showed evidence that some pigs were exposed to more than one avian influenza virus during their short lives. Because of the serological methods used in this study, the detection of multiple AIV antibodies in a given individual does not necessarily suggest co-infection (the exposure can be simultaneous or consecutive given the serological test cannot tell when the infection(s) happened). In most regions in Senegal, pigs are reared in backyard settings in low biosecurity conditions, where there is close intermingling of humans, chickens, ducks and pigs. Given the close interaction between the different species and the widespread outbreaks of the highly pathogenic avian influenza H5N1 virus, as was noted in Pout, Thiés region in 2020 [26], the risk of reassortment between avian A/H5N1 and A/H1N1pdm09 also detected in the swine population remains high, which may enhance replication and transmissibility competence compared to the parent H5N1 [71].

However, there are some limitations in the current active surveillance activities undertaken in Senegal. Firstly, samples are exclusively collected at the single pig slaughterhouse in Dakar, which could lead to a delay in detecting new emerging strains in the field and predominantly limits sampling to apparently healthy animals of market age. Furthermore, it would be interesting to carry out surveillance of influenza viruses in domestic poultry and wild birds in the country’s wetlands, which would allow a better analysis of the epidemiology of AIVs in Senegal and therefore help to better assess the risk of transmission to human. Secondly, only a small number of swIAVs were sequenced, which could lead to a lack of detection of reassortment events, and the absence of the characterization of H1N2 viruses also constitutes a weakness of this study. Indeed, an amplicon-based next-generation sequencing approach is ongoing to obtain complete genomes of H1N2.

## 5. Conclusions

In summary, the present study provides an overview of the epidemiology and genetic characteristics of influenza viruses circulating in the Senegalese pigs population, as well as serological evidence of avian influenza infection in this swine population. Globally, results of this active surveillance have shown a relatively high prevalence of influenza viruses in Senegalese pig herds, with a year-round circulation (peak of infection during the rainy season between June and September) of at least three influenza strains, including A/H1pdm09, A/H1N2 and A/H3N2. Molecular studies revealed multiple introductions of H1N1pdm09-like sublineage and human-like lineages that resembled pandemic A/H1N1pdm09 and seasonal A/H3N2 strains in human in Senegal, respectively. The occurrence of frequent amino acid mutations in both influenza A/H1N1pdm09 and A/H3N2 viruses support the need for whole genome sequencing, as well as analyzing virus evolution and gene mutations in a timely manner. Our study also revealed serological evidence of infection by at least four AIVs in pigs in Senegal, including A/H9N2, A/H5N1, A/H7N7 and A/H5N2, with a high seroprevalence rates.

Given the co-circulation of multiple subtypes of influenza viruses among Senegalese pigs, the potential exists for the emergence of new hybrid viruses of unpredictable zoonotic and pandemic potential in the future. Therefore, regular monitoring and frequent surveillance of influenza viruses in pigs and in markets that contain wild birds and live poultry are essential as part of an overall approach to the prevention and pandemic preparedness of animal and human influenza.

## Figures and Tables

**Figure 1 microorganisms-11-01961-f001:**
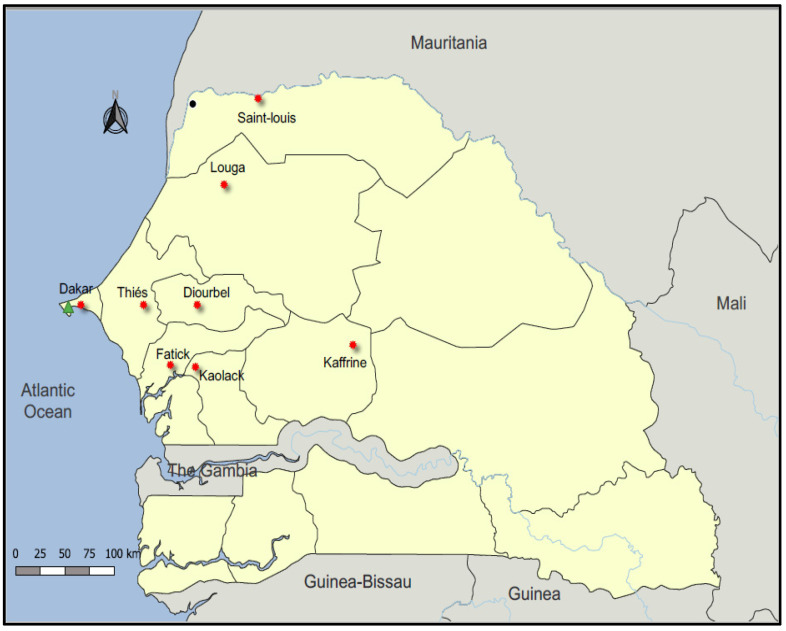
Map of Senegal showing origin of pigs from September 2018 to December 2019; orange shadowed dots indicate regions of pigs’ origin, and the Dakar pig slaughter house is indicated by a green triangle. The map was produced using QGIS software version 2.18.4 using public domain data obtained from natural Earth (http://www.naturalearthdata.com/, accessed on 21 March 2019).

**Figure 2 microorganisms-11-01961-f002:**
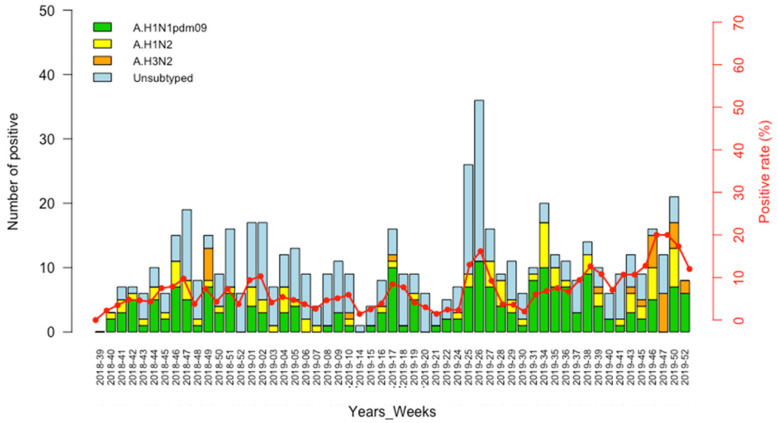
Weekly distribution of influenza subtypes in Senegalese pigs, September 2018–December 2019. Bars represent the proportions of IAV-positive cases for each epidemiological week and the curve represent the positivity rates.

**Figure 3 microorganisms-11-01961-f003:**
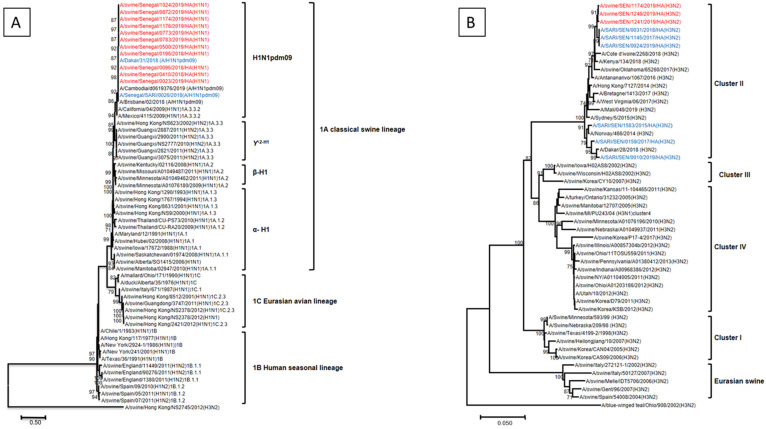
Phylogenetic trees of complete HA genes of H1 (**A**) and H3 (**B**) viruses circulating in swine and human population in Senegal (2018–2019). Trees were constructed using the maximum likelihood (ML) method, as implemented in MEGA 7 software. The robustness of the nodes was tested with 1000 bootstrap replications and bootstrap support values greater than 70 are shown at the nodes. Senegalese swine influenza A viruses are represented in red color while those in blue represent influenza A viruses circulating in the human population in Senegal.

**Figure 4 microorganisms-11-01961-f004:**
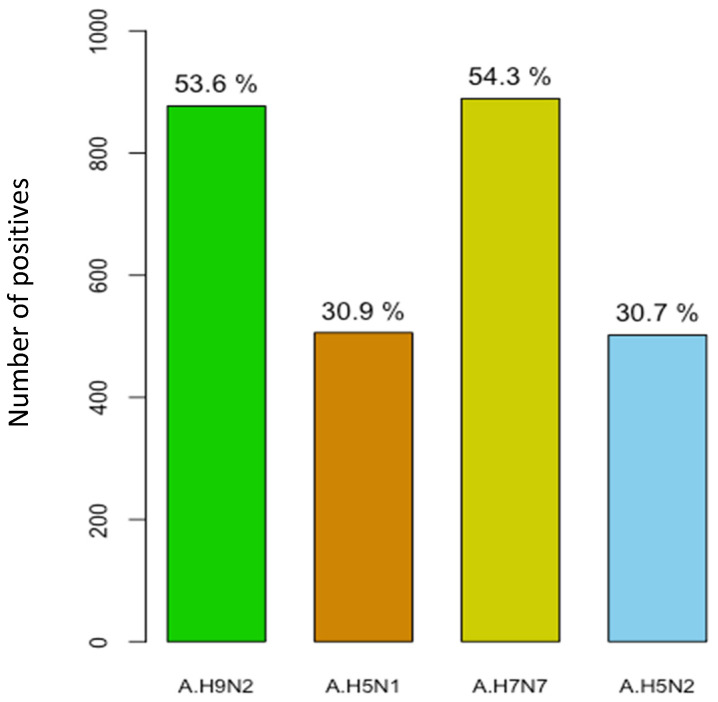
Distribution of AIV seropositive cases in pigs in Senegal, September 2018 to December 2019.

**Figure 5 microorganisms-11-01961-f005:**
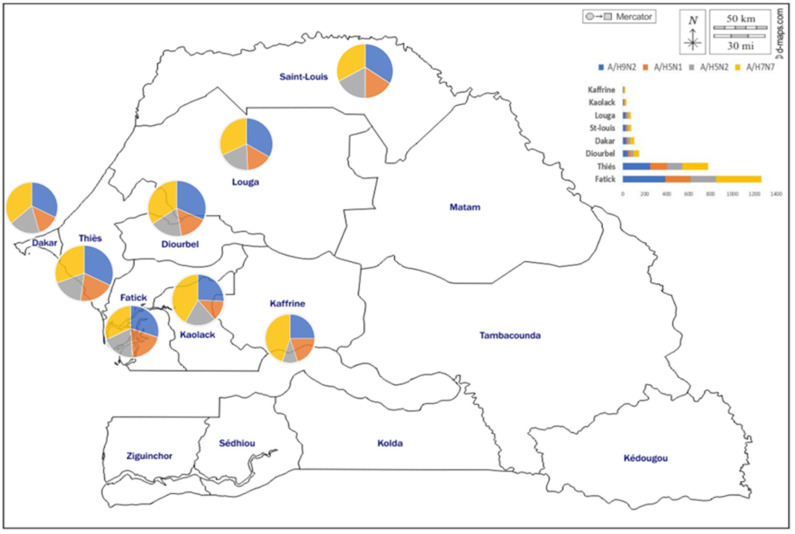
Geographical distribution of AIV seropositive cases in pigs in Senegal from September 2018 to December 2019.

**Table 1 microorganisms-11-01961-t001:** Detection rates of influenza A subtypes by region of origin in Senegalese swine herds, September 2018–December 2019.

	Tested	Positive Inf A	H and N Types				Detection Rates of Influenza A Subtypes			
Subtypes, n (%)	N (%)	N (%)	H1	N1	H3	N2	A/H1N1pdm09	A/H3N2	A/H1N2	Unsubtyped
Region, n (%)										
Dakar	69 (4.08)	19 (3.6)	6 (2.8)	6 (3.0)	2 (7.4)	4 (4.0)	6 (3.03)	2 (7.4)	2 (2.3)	11 (4.07)
Diourbel	109 (6.4)	41 (7.9)	21 (9.8)	18 (8.8)	0 (0.0)	10 (9.8)	18 (9.1)	0 (0)	10 (11.7)	20 (7.4)
Fatick	778 (46)	232 (44.6)	95 (44.4)	90 (44.1)	13 (48.1)	44 (43.1)	86 (43.4)	13 (48.1)	37 (43.5)	120 (44.44)
Kafrine	15 (0.9)	3 (0.57)	2 (0.9)	2 (1)	0 (0.0)	0 (0.0)	2 (1.01)	0 (0)	0 (0)	1 (0.37)
Kaolack	20 (1.2)	2 (0.38)	1 (0.5)	1 (0.5)	0 (0.0)	0 (0.0)	1 (0.5)	0 (0)	0 (0)	1 (0.37)
Louga	43 (2.5)	11 (2.1)	2 (0.9)	1 (0.5)	2 (7.4)	4 (4.0)	1 (0.5)	2 (7.4)	2 (2.3)	6 (2.2)
Saint louis	44 (2.6)	9 (1.7)	3 (1.4)	3 (1.5)	0 (0.0)	0 (0.0)	3 (1.5)	0 (0)	0 (0)	6 (2.2)
Thiés	449 (26.5)	153 (29.4)	62 (29.0)	58 (28.4)	7 (25.9)	30 (29.4)	57 (28.8)	7 (25.9)	24 (28.2)	83 (30.7)
Missing	164 (9.7)	50 (9.6)	22 (10.3)	25 (12.2)	3 (11.1)	10 (9.8)	24 (12.1)	3 (11.1)	10 (11.7)	22 (8.1)
Years, n (%)										
2018	410 (24.2)	113 (21.7)	49 (22.9)	47 (23.0)	5 (18.5)	24 (23.5)	47 (23.7)	5 (18.5)	19 (22.3)	56 (20.4)
2019	1281 (75.7)	407 (78.3)	165 (77.1)	157 (77.0)	22 (81.5)	78 (76.5)	151 (76.3)	22 (81.5)	66 (77.6)	219 (79.6)
Total	1691	520 (30.7)	214 (100)	204 (100)	27 (100)	102 (100)	198 (38.07)	27 (5.2)	85 (16.3)	275 (51.9)

**Table 2 microorganisms-11-01961-t002:** Genetic analysis of amino acid residues in the HA, NA, M2, PA and NS genes of Senegalese swine influenza, 2018–2019.

Virus	Subtype	Cleavage Site	HA						NA		M2					PA	NS	
			204	206	240	153	154	241	274	294	26	27	30	31	34	560	19	106
A/GuangdongMaonan/SWL1536/2019 ^a^	H1N1	PSIQSR/GLF	V	E	Q				H	N	L	V	A	N	G	P	M	M
A/swine/SEN/0096/2018 ^b^	H1N1	PSIQSR/GLF	D	Q	Q				H	N	L	V	A	N	G	P	M	M
A/swine/SEN/0196/2018 ^b^	H1N1	PSIQSR/GLF	D	Q	Q				H	N	L	V	A	N	G	S	L	M
A/swine/SEN/0410/2018 ^b^	H1N1	PSIQSR/GLF	D	Q	Q				H	N	L	V	A	N	G	P	M	M
A/swine/SEN/0023/2019 ^b^	H1N1	PSIQSR/GLF	D	Q	Q				H	N	L	V	A	N	G	P	M	M
A/swine/SEN/0500/2019 ^b^	H1N1	PSIQSR/GLF	D	Q	Q				H	N	L	V	A	N	G	P	M	M
A/swine/SEN/0773/2019 ^b^	H1N1	PSIQSR/GLF	D	Q	Q				H	N	L	V	A	N	G	P	M	M
A/swine/SEN/0783/2019 ^b^	H1N1	PSIQSR/GLF	D	Q	Q				H	N	L	V	A	N	G	P	M	M
A/swine/SEN/0872/2019 ^b^	H1N1	PSIQSR/GLF	D	Q	Q				H	N	L	V	A	N	G	P	M	M
A/swine/SEN/1024/2019 ^b^	H1N1	PSIQSR/GLF	D	Q	Q				H	N	L	V	A	N	G	P	M	M
A/swine/SEN/1174/2019 ^b^	H1N1	PSIQSR/GLF	D	Q	Q				H	N	L	V	A	N	G	P	M	V
A/swine/SEN/1176/2019 ^b^	H1N1	PSIQSR/GLF	D	Q	Q				H	N	L	V	A	N	G	P	M	M
A/Dakar/31/2018 ^c^	H1N1	PSIQSR/GLF	D	Q					H	N	L	V	A	N	G			
A/Dakar/26/2018 ^c^	H1N1	PSIQSR/GLF			Q				H	N	L	V	A	N	G			
A/HongKong/2671/2019 ^a^	H3N2	PEKQTR/GI				F	S	N	H	N	L	V	A	N	G	P	M	M
A/swine/SEN/1174/2019 ^b^	H3N2	PEKQTR/GI				S	A	D	H	N	L	V	A	N	G	P	M	M
A/swine/SEN/1241/2019 ^b^	H3N2	PEKQTR/GI				S	A	D	H	N	L	V	A	N	G	P	M	M
A/swine/SEN/1249/2019 ^b^	H3N2	PEKQTR/GI				S	A	D	H	N	L	V	A	N	G	P	M	M
A/Senegal/0024/2019 ^c^	H3N2	PEKQTR/GI				S	A	D	H	N	L	V	A	N	G			
A/Senegal/0010/2019 ^c^	H3N2	PEKQTR/GI		D					H	N	L	I	A	N	G			

^a^ Reference viruses, ^b^ Influenza A viruses detected in Senegalese pigs, ^c^ Influenza A viruses circulating in the human population in Senegal.

**Table 3 microorganisms-11-01961-t003:** HI seroprevalence of AIVs (H9N2, H5N1, H7N7, H5N2) in pigs imported from different regions of Senegal, 2018–2019.

		HI Positivity Rates to Different AIV Antigens							
Regions	No. of Sera (%)	H9N2 (%)	95%CI	H5N1 (%)	95%CI	H7N7 (%)	95%CI	H5N2 (%)	95%CI
Dakar	67 (4.1)	33 (3.8)	2.0–5.0	14 (2.8)	1.0–5.0	37 (4.2)	3.0–6.0	20 (4.0)	2.0–6.0
Diourbel	106 (6.5)	47 (5.3)	4.0–7.0	23 (4.5)	3.0–7.0	51 (5.7)	4.0–7.0	27 (5.4)	3.6–7.8
Fatick	750 (45.8)	388 (44.2)	40.0–47.0	232 (45.8)	41.0–50.0	414 (46.6)	43.0–50.0	229 (45.6)	41.0–50.0
Kaffrine	15 (0.9)	5 (0.6)	0.2–1.0	4 (0.8)	0.2–2.0	9 (1.0)	0.5–2.0	2 (0.4)	0.07–1.6
Kaolack	19 (1.2)	8 (0.9)	0.4–2.0	4 (0.8)	0.2–2.0	13 (1.5)	0.8–2.5	6 (1.2)	0.48–2.7
Louga	42 (2.6)	25 (2.8)	2.0–4.0	12 (2.4)	1.2–4.2	24 (2.7)	1.7–4.0	14 (2.8)	1.5–4.7
Saint-louis	42 (2.6)	26 (3.0)	2.0–4.0	13 (2.6)	1.4–4.4	25 (2.8)	1.8–4.1	14 (2.8)	1.5–4.7
Thiés	440 (26.9)	249 (28.4)	25.0–31.0	157 (31.0)	27.0–35.0	238 (26.8)	23.9–29.8	135 (26.9)	23.0–31.0
Missing	155 (9.5)	96 (11.0)	9.0–13.0	47 (9.3)	7.0–12.0	78 (8.8)	7.0–10.8	55 (11.0)	8.0–14.0
Years									
2018	394 (24.1)	268 (30.5)	27.0–34.0	210 (41.5)	37.0–45.0	285 (32.0)	29.0–35.0	215 (42.8)	38.0–47.0
2019	1242 (75.9)	609 (69.4)	66.0–72.0	296 (58.5)	54.0–63.0	604 (68.0)	64.7–70.9	287 (57.2)	53.0–61.0
Total	1636 (100)	877 (53.6)	51.0–56.0	506 (31.0)	29.0–33.0	889 (54.3)	51.8–56.7	502 (30.7)	28.0–33.0

## Data Availability

The swIAV sequences generated in this study have been deposited in GISAID database under accession numbers EPI_ISL_18017683, EPI_ISL_18017032, EPI_ISL_18017793, EPI_ISL_18018393, EPI_ISL_18018905, EPI_ISL_18019301, EPI_ISL_18019813, EPI_ISL_18020349, EPI_ISL_18020894, EPI_ISL_18021622, EPI_ISL_18022138, EPI_ISL_18022699, EPI_ISL_18023460, EPI_ISL_18023925.

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
