# Peer review of "Influenza A Virus in Pigs in Senegal and Risk Assessment of Avian Influenza Virus (AIV) Emergence and Transmission to Human"

_microorganisms, 2023, doi:10.3390/microorganisms11081961_

Round 1
Reviewer 1 Report
The manuscript by Jallow et al. describes an active surveillance project whereby samples collected from a pig slaughterhouse in Senegal were assessed for the presence of influenza A viruses by molecular and serological methods.
Minor comments:
Line42-43: The number of viral proteins encoded in the influenza A viruses can be variable, therefore this statement is unclear without further clarification. As the authors then go on to describe all influenza viruses, including Influenza C and D which have a different number of gene segments (and therefore a different number of encoded viral proteins) compared to Influenza A and B, the line 'encoding up to eleven different viral proteins' should be removed.
Line 47: 'antigenetic' - do the authors mean antigenic or genetic? This should be corrected.
Line 48: The author should clarify that annual seasonal epidemics and mortality are related to human influenza viruses, not veterinary influenza viruses.
Line 59: The list of host species provided by the authors is not exhaustive and should be acknowledged as such in the text.
Line 59: The authors should define the term reassort to explain how novel influenza viruses can be generated by this mechanism.
Line 63: 'pandemics' should not be plural.
Line 63: The authors state 'This process…' but it is not clear what process is being referred to, the authors should improve the clarity of this section. Also, the authors should also refer to antigenic drift as this is a mechanism through which influenza viruses can diversify in addition to antigenic shift.
Lines 67-68: The SA initialisation in alpha-2,3-SA and alpha-2,6-SA has not been defined, the authors should correct this.
Lines 67-70: Attachment of different influenza A viruses to sialic acid residues in the respiratory tract is not sufficient for reassortment to occur, infection of the same cell is also required. The authors should correct this statement.
Line 88: The HPAIV initialisation has not been previously defined. This should be done at the first mention in line 85.
Line 89: The AIV initialisation has not been defined, the authors should correct this.
Line 109: The IPD initialisation has not been defined in the manuscript text, the authors should correct this.
Line 124: The authors should clarify that RNA was extracted from the nasal swab VTM fluid, not the swabs themselves.
Line 129: The RT-qPCR initialisation has not been defined, the authors should correct this.
Line 132: Reference 28 does not provide sufficient details regarding the primer and probe sequences of the RT-qPCR assays used in this study. These sequences should be provided in the supplementary materials for all RT-qPCR assays used in the study to enable the reader to fully understand the outcomes of the testing described in the manuscript.
Line 138: The initial concentration of the primers and probes should be provided to enable reproducibility of the testing regimen.
Lines 141-142: The authors should provide details of the positive and negative controls, as well as how they were used to define a valid test for the study.
Lines 155-172: It is not clear how the cDNA used for the multi-segment RT-PCR was generated. If this is described in Lines 144-154, then this should be clarified in the text. If this is the case though, then the RT-PCR procedure described in Lines 155-172 is actually just a PCR and not an RT-PCR. The authors should correct this section to improve clarity of the procedures used.
Line 171: The P2M initialisation has not been defined, the authors should correct this.
Line 242: The (3/4) description is irrelevant as the actual percentage and proportion are provided in line 243.
Line 252: The authors should clarify that it was the IAV-positive samples that were subtyped, not the IAV itself.
Line 263: The initialisation SIV has not been defined and another initialisation was used previously (swIAV). The authors should only use a single initialisation to denote swine influenza viruses throughout the manuscript for consistency.
Line 295: HA3 should be replaced with HA.
Figure 3: In part A, the colloquial names (e.g., H1N1pdm09, triple reassortant etc.) should be replaced with, or provided in addition to the ratified nomenclature described by Anderson et al. 2016 (https://pubmed.ncbi.nlm.nih.gov/27981236/). In part B, one of the Senegalese swine samples is incorrectly labelled as H1N1, this should be corrected.
Lines 305-326: Are these amino acid substitutions observed in the human Senegalese sequences as well? As the swine sequences are more related to human sequences from Senegal, these would represent much more appropriate references than those currently used in the manuscript from China and Australia.
Figure 4: The y-axis is titled as seropositivity rates, whereas the axis ticks would suggest it is actually showing the number of positives. The authors should correct this.
Lines 465-466: This is the first definition of the PA segment, this and other definitions should be provided much earlier in the manuscript at the first use of the segment nomenclature.
Major Comments:
Lines 173-181: The details of how the sequencing libraries were prepared lacks any details of the kits/reagents used and therefore prevents reproducibility of the procedure. The authors must include full details of all kits/reagents used.
Lines 173-181: Details for how the authors processed the raw sequencing reads to generate the consensus sequences are completely lacking. The authors must include full details for how this was achieved to enable reproducibility of the work described in the manuscript.
Lines 183-187: The timeframe covered by the sequence data downloaded from NCBI/GISAID does not cover the entire length of the study. The authors should explain why this is the case, and consider the impact of not including global swine influenza sequences that were generated after the study on the outcomes of the analyses performed.
Lines 195-199: The authors only tested the swine sera using avian influenza viral antigens and did not include human or swine H1 and H3 viral antigens. HI assays using such H1/H3 antigens would be more informative to the reader and would relate to the molecular testing described as opposed to just using avian influenza viruses. The inclusion of HI testing of the sera against avian influenza is informative to the reader and will assist with furthering our understanding of the ability of such viruses to potentially infect swine, however, such testing should only be performed in addition to, and not in place of serological testing for H1/H3 influenza viruses, and should be accompanied with molecular testing for these avian influenza viruses.
Lines 208-210: The WHO recommendations describe the use of 8HA units/50 uL (4HA units/25 uL) of viral antigen for HI assays, therefore the reference does not support the procedure used, and is not commonplace in the field. Is this a typo? If not, the authors should describe why this deviation has been used, or provide a more accurate reference for the method.
Lines 257-259 and Table 1: In Table 1 the authors should provide the results of the individual H-type and N-type RTR-qPCR separately, along with a column that then identifies the aggregate subtype. Without this information it is not possible to fully identify the subtypes in the potential co-infection samples. This is also confounded by a lack of detail provided in the methods regarding the assays used for subtyping the samples. If there are co-infections this could also confound the output of the NGS results, and should be clarified whether such samples were used in the phylogenetic analysis (lines 278-299).
Lines 263-265: The description that there was a higher prevalence of swine influenza in 2019 versus 2018, is confounded by the fact that the surveillance program started only in September 2018, and that therefore the majority of samples were collected in 2019. When considering the proportion of positives from 2018 (113 out of 410 samples tested; 27.6%) and 2019 (407 out of 1281; 31.8%) there is little difference (4.2%), therefore this statement is incorrect and misleading and must be corrected.
Line 271-272: This statement is incorrect as whilst the peak of A/H3N2 detections occurred between week 43-52 of 2019, A/H3N2 was also detected in weeks 10,17 and 39 of 2019 and should be corrected.
Data availability: The manuscript details that the study data is available upon request. However, this is not the accepted process for genomic data, which should be made publicly available (via NCBI or GISAID) once the manuscript is published. This and all sequence accession numbers should be detailed within the manuscript.
Overall the quality of English language used throughout the manuscript is acceptable, however, there are several instances where there are grammatical mistakes or typographical errors. The authors should seek to rectify these to improve the readability of the article.
Author Response
The manuscript by Jallow et al. describes an active surveillance project whereby samples collected from a pig slaughterhouse in Senegal were assessed for the presence of influenza A viruses by molecular and serological methods.
Thank you very much for the great interest you have shown for this article and for your insightful comments and suggestions which will undoubtedly help to improve the quality of this paper.
Minor comments:
Line42-43: The number of viral proteins encoded in the influenza A viruses can be variable, therefore this statement is unclear without further clarification. As the authors then go on to describe all influenza viruses, including Influenza C and D which have a different number of gene segments (and therefore a different number of encoded viral proteins) compared to Influenza A and B, the line 'encoding up to eleven different viral proteins' should be removed.
Thank you for highlighting this. Indeed, the line ‘encoding up to eleven different viral proteins` is removed as suggested.
Line 47: 'antigenetic' - do the authors mean antigenic or genetic? This should be corrected.
The reviewer is right, we meant to write antigenic instead of antigenetic, it is now corrected.
Line 48: The author should clarify that annual seasonal epidemics and mortality are related to human influenza viruses, not veterinary influenza viruses.
Thank you for the suggestion. We have rephrased the sentence.

Reviewer 2 Report
Influenza A Virus in Pigs in Senegal and Risk Assessment of Avian Influenza Virus (AIV) Emergence and Transmission to Human
Authors describe an active influenza surveillance in the single pig slaughterhouse in Dakar to investigated the epidemiology and genetic characteristics of Influenza A viruses and serological evidence of Avian Influenza viruses in pigs.
In my opinion the title does not represent the provided information in the article. The serological investigation in pigs is accurately presented, but there is no epidemiological evidence about risk assessment in human population.
Section Materials and Methods.
Line 132, describe more precisely the used CDC protocol, as you did about other used protocols.
You tested swab samples for H1N2, H3N2, and H1N1pdm, there was an outbreak of H5Nq in birds… Why did not include H5N1? Your aim is to monitor circulating influenza strains in animals and to estimate the risk for human health.
Lines 195-198, why you used exactly these viruses H7N7, H9N2, H5N2? You mentioned in the Introduction that Senegal faced a large outbreak of H5N1. It makes sense, why you tested the sera for antibodies against H5N1, but why you choose others. You did not include swine viruses, such as H1N1, H3N2, N1N2, why? The description of HI method is very detailed. It could be shortened.
Figure 3, very nice phylogenetic trees, but write a legend which viruses are in blue and which are in red
Section Discussion.
As you wrote, subtypes H1N1 and H3N2 are circulating in the human population in Senegal, but the data are not shown. Table 2 very good present the data. If you could match it with people's information and do analysis, it will be better.
Did you do epidemiological investigation of positive sera samples of the pigs? Are there poultry farms near the pig farms, or all the samples are from backyard animals? It is good to see what is the connections between poultry and pigs? For me, the high seroprevalence rate of AIV is somewhat surprising.
In conclusion, I will say that the provided data is interesting, but the risk assessment on human health should be better specified and described or the title should be revised.
Author Response
Influenza A Virus in Pigs in Senegal and Risk Assessment of Avian Influenza Virus (AIV) Emergence and Transmission to Human
Authors describe an active influenza surveillance in the single pig slaughterhouse in Dakar to investigated the epidemiology and genetic characteristics of Influenza A viruses and serological evidence of Avian Influenza viruses in pigs.
In my opinion the title does not represent the provided information in the article. The serological investigation in pigs is accurately presented, but there is no epidemiological evidence about risk assessment in human population.
Thank you for the great interest you showed for this article, and especially for your insightful comments and suggestions which will undoubtedly help to improve the quality of this paper.
Section Materials and Methods.
Line 132, describe more precisely the used CDC protocol, as you did about other used protocols.
Thank you we have now described the full details of the CDC protocol as follows:
“RT-qPCR was performed to initially screen for all IAVs using specific primers and probe targeting a conserved region of the influenza matrix gene and provided by the CDC International Reagent Resources (IRR) using an ABI 7500 device, according to the CDC procedure [28]. Briefly, tests are performed in singleplex using The AgPath-ID™ One-Step RT-PCR kit (Ambion, Foster City, CA, USA). For each sample, real time PCR was carried out in a total reaction volume of 25 μl consisting of 5 μl Nuclease-free water, 0.5 μl of each primer, 0.5 μl of probe, 12.5 μl of 2X RT-PCR Buffer, 1 μl 25X RT-PCR Enzyme Mix and 5 μl of RNA template under the following cycling conditions: reverse transcription step of 15 min at 50°C, initial denaturation step of 3 min at 95°C, followed by 45 PCR cycles of 15 sec at 95°C and 30 sec at 55°C. For each test, controls (positive and negative) were included for validation. Subsequently, all M gene-positive samples (Ct-values < 36) were subtyped by RT-qPCR assays using specific primers and probe that specifically amplify and discriminate the HA and NA genes from the different lineages of swine IAVs known to be enzootic in pig herds (H1N2, H3N2, and H1N1pdm). The same condition as the initial screening was used for subtyping.”
You tested swab samples for H1N2, H3N2, and H1N1pdm, there was an outbreak of H5Nq in birds… Why did not include H5N1? Your aim is to monitor circulating influenza strains in animals and to estimate the risk for human health.
Thank you. Swab samples from pigs were tested for all influenza A viruses (including H5N1), we did not include H5N1 in the subtyping because there is no suspicion of these viruses in our preliminary results (all flu A positive were subtyped H1N2, H3N2, and H1N1pdm). Just to remind that the present study aimed to monitor circulating influenza viruses at the human/pigs interface and assess the risk of transmission to human. However, in perspective of this study we want to conduct an active surveillance of influenza viruses in domestic poultry and wild birds in the country's wetlands to better analysis the epidemiology of AIVs and therefore to better assess the risk of transmission to human.
Lines 195-198, why you used exactly these viruses H7N7, H9N2, H5N2? You mentioned in the Introduction that Senegal faced a large outbreak of H5N1. It makes sense, why you tested the sera for antibodies against H5N1, but why you choose others. You did not include swine viruses, such as H1N1, H3N2, N1N2, why? The description of HI method is very detailed. It could be shortened.
We used these viruses in addition to H5N1 because they are among the most common avian influenza viruses and sporadic human cases of these viruses have been reported in several countries around the world. This is why we wanted to determine the seroprevalence of these viruses in pigs in Senegal. We did not include these swine viruses for the serology as we tested them by RT-PCR. We admit that the HI method is very detailed but we felt that the assay being relatively too heavy will only be beneficial for the readers.
Figure 3, very nice phylogenetic trees, but write a legend which viruses are in blue and which are in red
Thank you for this feedback, we have now updated the legend in the phylogenetic trees.
Section Discussion.
As you wrote, subtypes H1N1 and H3N2 are circulating in the human population in Senegal, but the data are not shown. Table 2 very good present the data. If you could match it with people's information and do analysis, it will be better.
Thank you for this suggestion, we have now updated Table 2 to include information of Influenza A viruses circulating in the human population in Senegal.
Did you do epidemiological investigation of positive sera samples of the pigs? Are there poultry farms near the pig farms, or all the samples are from backyard animals? It is good to see what is the connections between poultry and pigs? For me, the high seroprevalence rate of AIV is somewhat surprising.
No, we did not do epidemiological investigation of positive sera samples of the pigs, as this is beyond our competence. In Senegal, epidemiological investigation in animal is under the responsibility of the ministry of Livestock. All the samples tested in this study were from pigs reared in backyard settings in low biosecurity conditions, where there is close intermingling of humans, chickens, ducks and pigs. We agree with the reviewer, it would be interesting to see what is the connections between poultry and pigs, which is why in perspective of this study we want to conduct an active surveillance of influenza viruses in domestic poultry and wild birds in the country's wetlands to better analysis the epidemiology of AIVs in Senegal and therefore to better assess the risk of transmission to human.
In conclusion, I will say that the provided data is interesting, but the risk assessment on human health should be better specified and described or the title should be revised.
Reviewer 3 Report
The wide range of hosts provides influenza A viruses greater chances of genetic re-assortment, leading to the emergence of zoonotic strains and occasional pandemics that have a severe impact on human life. Therefore, monitoring circulating strains of influenza in animals and evaluating their potential to cross the species barrier and cause human infections is important for good pandemic preparedness. In this study, Jallow et al. described the epidemiology and genetic characteristics of the influenza viruses circulating in the Senegalese pig population as well as serological evidence of avian influenza infection in these swine population. The study is well design and I completely support to purplish it after minor modifications.
Line 18: Use IAV's full name, this is used for the first time and for all other abbreviations used for the first time in the manuscript.
Line 22-23: Confusing statement, please explain clearly what the four types of AIV are?
Line 109: What is IPD abbreviated for?
Line 122: Please replace swine influenza viruses (swIAVs) into swine influenza A viruses (swIAVs)
Line 123: The swab samples were not propagated in SPF chicken eggs? Did you quantify the extracted RNA and how much amount did you get for PCR and later for cDNA synthesis and NGS?
Line 251-254: Confusing to readers, a total of 520 out of 1691 samples were positive for IAV. The detection rate of virus A/H1N1pdm09 was 38.1% (198/520), followed by viruses A/H1N2 and A/H3N2 with detection rates of 16.3%. (85/520) and 5.2% (27/520), these are for 310 (57.61%) positive samples and where are another 210 (40.38%) positive samples?
Line 255-257: They explained at line 132 in Materials and Methods that all M gene positive samples (Ct values ​​< 36) were subsequently subtyped using RT-qPCR assays. However, the subtype could not be determined for the 270 positive samples as they had M gene Ct values ​​>33, indicating an apparently very low viral load. Please explain the criteria used to classify a sample as positive and then subtype it clearly.
Line 255: Please correct the number 270 mentioned in the text, but the last column of table 1 revealed 275
Author Response
Comments and Suggestions for Authors
The wide range of hosts provides influenza A viruses greater chances of genetic re-assortment, leading to the emergence of zoonotic strains and occasional pandemics that have a severe impact on human life. Therefore, monitoring circulating strains of influenza in animals and evaluating their potential to cross the species barrier and cause human infections is important for good pandemic preparedness. In this study, Jallow et al. described the epidemiology and genetic characteristics of the influenza viruses circulating in the Senegalese pig population as well as serological evidence of avian influenza infection in these swine population. The study is well design and I completely support to purplish it after minor modifications.
Thank you for the global positive appreciation you did regarding this work, and also for the comments and suggestions which will undoubtedly help to improve the quality of this paper.
Line 18: Use IAV's full name, this is used for the first time and for all other abbreviations used for the first time in the manuscript.
Thank you for the suggestion, the full name of IAV is now written in this version.
Line 22-23: Confusing statement, please explain clearly what the four types of AIV are?
We admitted that sentences from lines 22 to 23 is not clear. We have revised this sentence, and this now reads:
“Serum samples were tested by HI assay for the detection of antibodies recognizing four AIVs, including H9N2, H5N1, H7N7 and H5N2”.
Line 109: What is IPD abbreviated for?
IPD is the abbreviation for Institut Pasteur de Dakar
Line 122: Please replace swine influenza viruses (swIAVs) into swine influenza A viruses (swIAVs)
Correction is done.
Line 123: The swab samples were not propagated in SPF chicken eggs? Did you quantify the extracted RNA and how much amount did you get for PCR and later for cDNA synthesis and NGS?
Indeed, the swab samples were not propagated in SPF chicken eggs nor did we quantify the extracted RNA. However, for cDNA synthesis and NGS, positive samples with Ct values ranging from 16 to 20 during the PCR were selected (as lowest Ct-values are correlated with important viral load) but also in taking into account the temporal distribution.
Line 251-254: Confusing to readers, a total of 520 out of 1691 samples were positive for IAV. The detection rate of virus A/H1N1pdm09 was 38.1% (198/520), followed by viruses A/H1N2 and A/H3N2 with detection rates of 16.3%. (85/520) and 5.2% (27/520), these are for 310 (57.61%) positive samples and where are another 210 (40.38%) positive samples?
Thank you for these remarks, indeed, out of the 1691 samples tested, 520 samples were positive. However, it should be noted that among the 310 positive samples mentioned by the reviewer, there are co-infections of several subtypes which are therefore counted separately for each coinfecting virus. Of the 520 positive samples detected in this work, single infection by A/H1N1pdm09 was encountered in 126 samples, mono-infections by A/H1N2 and A/H3N2 in 14 and 18 samples respectively. Co-infection H1N1pdm09/H1N2 was noted in 70 specimens, co-infection H1N1pdm09/H3N2 in 8 cases and co-infection H1N2/H3N2 in 7 cases, giving a total of 243 positive samples. The remaining 277 positive samples represents unsubtyped samples (275 single infections and 2 co-infections).
Line 255-257: They explained at line 132 in Materials and Methods that all M gene positive samples (Ct values ​​< 36) were subsequently subtyped using RT-qPCR assays. However, the subtype could not be determined for the 270 positive samples as they had M gene Ct values ​​>33, indicating an apparently very low viral load. Please explain the criteria used to classify a sample as positive and then subtype it clearly.
Thank you for highlighting this. In this study, we consider a sample positive when it is detected with a Ct-value < 36. Subsequently a subtyping is carried out on all the positive samples but we noticed that beyond a Ct value >33 the sample fails to be subtyped.
Line 255: Please correct the number 270 mentioned in the text, but the last column of table 1 revealed 275
Indeed, that was a mistake. The number is corrected.
Reviewer 4 Report
In the manuscript entitled, " Influenza A Virus in Pigs in Senegal and Risk Assessement of Avian Influenza Virus (AIV) Emergence and Transmission to Human", newly sequences in a district. The manuscript is generally well-addressed and well-written. I have some minor comments:
Line 18: please clarify abbreviation of IAVs.
Line 195: the sentence “HI assay for the presence or not of antibodies…” please rewrite it.
Line 209: Why use 8 HA units/25 μl of reference antigens in the HI test? It should be 8 HA units/50μl (4HA units/25 μl )?
Line 278: title: Phylogenetic analysis of IVA strains...." Please correct IVA.
Please revise the references and follow the journal guidelines.
Minor edits
Author Response
Comments and Suggestions for Authors
In the manuscript entitled, " Influenza A Virus in Pigs in Senegal and Risk Assessement of Avian Influenza Virus (AIV) Emergence and Transmission to Human", newly sequences in a district. The manuscript is generally well-addressed and well-written. I have some minor comments:
Thank you very much for your time and interest in this work, and for the very useful comments which will help us to strengthen the paper.
Line 18: please clarify abbreviation of IAVs.
Done as requested
Line 195: the sentence “HI assay for the presence or not of antibodies…” please rewrite it.
The sentence has been rephrased as suggested by the reviewer.
Line 209: Why use 8 HA units/25 μl of reference antigens in the HI test? It should be 8 HA units/50μl (4HA units/25 μl)?
The reviewer is right; it was a mistake, we used 8 HA units/50μl. Correction has been made in this version and thank you for drawing our attention on this mistake.
Line 278: title: Phylogenetic analysis of IVA strains...." Please correct IVA.
Correction made as suggested
Please revise the references and follow the journal guidelines.
All references have been checked and revised when needed
Round 2
Reviewer 1 Report
Minor Comments
Line 19 - AIV should be defined in the abstract upon first usage. For the main text this has been done in line 94-95.
Line 90 - Thank you for defining the initialisation of HPAIV, one small correction though - it should be high pathogenicity not highly pathogenic.
Major Comments
Data availability - thank you for addressing the prior comment regarding listing the accession numbers in the manuscript. However, this cannot just be removed from the manuscript and the accession number for all genome sequences generated in this study should be included in the manuscript.
Minor review of the grammar required throughout to ensure consistency where sections have been added.
Author Response
Comments and Suggestions for Authors
Thank you once again for your time and the positive evaluation you did regarding this work, and also for the comments and suggestions which will help to improve the quality of this paper.
Minor Comments
Line 19 - AIV should be defined in the abstract upon first usage. For the main text this has been done in line 94-95.
Thank you for the feedback. AIV has been defined as suggested.
Line 90 - Thank you for defining the initialisation of HPAIV, one small correction though - it should be high pathogenicity not highly pathogenic.
Thank you. Corrected as suggested.
Major Comments
Data availability - thank you for addressing the prior comment regarding listing the accession numbers in the manuscript. However, this cannot just be removed from the manuscript and the accession number for all genome sequences generated in this study should be included in the manuscript.
Thank you once again for pointing this out. We have now deposited the sequences generated in this study in GISAID database under accession numbers EPI_ISL_18017683, EPI_ISL_18017032, EPI_ISL_18017793, EPI_ISL_18018393, EPI_ISL_18018905, EPI_ISL_18019301, EPI_ISL_18019813, EPI_ISL_18020349, EPI_ISL_18020894, EPI_ISL_18021622, EPI_ISL_18022138, EPI_ISL_18022699, EPI_ISL_18023460, EPI_ISL_18023925
Comments on the Quality of English Language
Minor review of the grammar required throughout to ensure consistency where sections have been added.
Revised